# HLA-G 14bp Ins/Del Polymorphism, Plasma Level of Soluble HLA-G, and Association with IL-6/IL-10 Ratio and Survival of Glioma Patients

**DOI:** 10.3390/diagnostics12051099

**Published:** 2022-04-27

**Authors:** Maria Bucova, Kristina Kluckova, Jan Kozak, Boris Rychly, Magda Suchankova, Marian Svajdler, Viktor Matejcik, Juraj Steno, Eszter Zsemlye, Vladimira Durmanova

**Affiliations:** 1Faculty of Medicine, Institute of Immunology, Comenius University, 813 72 Bratislava, Slovakia; kristina.kluckova@fmed.uniba.sk (K.K.); magdasuchankova@fmed.uniba.sk (M.S.); eszter.zsemlye@fmed.uniba.sk (E.Z.); vladimira.durmanova@fmed.uniba.sk (V.D.); 2Department of Neurosurgery, Faculty of Medicine, Comenius University and University Hospital, 833 05 Bratislava, Slovakia; kozakjan@hotmail.com (J.K.); viktor.matejcik@fmed.uniba.sk (V.M.); juraj.steno@fmed.uniba.sk (J.S.); 3Alpha Medical, Ltd., 841 01 Bratislava, Slovakia; rychly.boris@alphamedical.sk; 4Cytopathos Ltd., 831 03 Bratislava, Slovakia; svajdler@yahoo.com; 5Sikl’s Department of Pathology, the Faculty of Medicine and Faculty Hospital in Pilsen, Charles University, 306 05 Pilsen, Czech Republic

**Keywords:** glioma, HLA-G, IL-6, polymorphism, prognosis, survival

## Abstract

HLA-G is an immune checkpoint molecule with immunosuppressive and anti-inflammatory activities, and its expression and level of its soluble form (sHLA-G) may play an important role in tumor prognosis. The HLA-G 14bp ins/del polymorphism and the plasma level of soluble HLA-G (sHLA-G) were investigated by a polymerase chain reaction and ELISA, respectively, in 59 glioma patients. A significantly higher proportion of glioma patients had the 14 nt insert in both homozygous and heterozygous states compared to the control group. Glioma patients also had higher plasma levels of sHLA-G. Patients with methylated MGMT promoters had lower levels of sHLA-G than those with unmethylated MGMT promoters. The level of sHLA-G negatively correlated with the overall survival of patients. Glioblastoma patients who survived more than one year after diagnosis had lower levels of sHLA-G than those surviving less than one year. Patients with sHLA-G levels below the cut-off value of 40 U/mL survived significantly longer than patients with sHLA-G levels above 40 U/mL. The levels of sHLA-G were also negatively correlated with the level of IL-6 (*p* = 0.0004) and positively with IL-10/IL-6 (*p* = 0.046). Conclusion: The presence of the 14 nt insert in both homozygous and heterozygous states of the HLA-G 14bp ins/del polymorphism is more frequent in glioma patients and the elevated plasma levels of sHLA-G are negatively associated with their survival.

## 1. Introduction

The HLA (human leukocyte antigen) system plays a key role in the development of immune tolerance and has an impact on both innate and adaptive immune responses. HLA-G is one of the non-classical HLA class I molecules of the major histocompatibility complex with well-characterized immunomodulatory activities [1]. Contrary to the classical HLA antigens, HLA-G is characterized by a low polymorphism and high tolerogenic functions. It is coded by a gene located on the short arm of the sixth chromosome (6p21.3) and encompasses at least four membrane-bound (mHLA-G: HLA-G1, HLA-G2, HLA-G3, and HLA-G4) and three soluble (sHLA-G: HLA-G5, HLA-G6, and HLA-G7) isoforms resulting from alternative splicing of its primary mRNA [2,3,4,5]. The HLA-G gene consists of eight exons; however, exon 8 remains untranslated due to the presence of a stop codon in exon 7. The non-translated region of exon 8 is termed the “3′ untranslated region (3′UTR)” [6]. Eighteen SNPs, a 14bp insertion/deletion, and forty-four haplotypes have been identified in the 3′UTR region [1], which are known to influence the translation of HLA-G proteins through either a reduced transcription, mRNA stability, or aberrant alternative splicing.

Under physiological conditions, HLA-G expression is highly tissue-restricted. It modulates the immune system activity in tissues where physiological immune tolerance is necessary—e.g., in the cytotrophoblast at the maternal−fetal interface [7] and in adults, in immune-privileged organs, including the cornea, thymus, pancreatic islets, endothelial cell precursors, and erythroblasts [2]. HLA-G plays a major role in protecting both the semi-allogenic tissues of the fetus from the maternal immune system and immune-privileged organs by creating a tolerogenic microenvironment.

HLA-G is a tolerogenic molecule that inhibits effector cells or generates regulatory subtypes that broadly regulate the immune system—both innate and adaptive immune responses and inflammation [8]. It is involved in maintaining tolerance in autoimmune, inflammatory diseases [9,10,11], and after transplantation [12,13,14,15], and its immunosuppressive and anti-inflammatory functions have been well recognized [16]. HLA-G also potentiates the immune escape in cancer and infectious diseases [17,18].

Many studies previously described the neo-expression of HLA-G in different types of cancer cells, and its correlation with histological grade [19] bad prognosis [20,21], tumor metastasis, and poor survival [22]. HLA-G plays a great role in creating a tolerogenic tumor microenvironment.

For proper anti-tumor immunity, both a well-functioning innate and adaptive Th1 immunity are required. However, tumor cells have developed various direct and indirect mechanisms impairing the functions of immune cells in the tumor microenvironment for their survival. Aberrant induction of HLA-G expression in malignant cells represents one of the key factors that contribute to tumor immune escape and progression. HLA-G with its immunosuppressive functions hampers the anti-tumor immunity and potentiates the cancer growth and its metastatic process. It plays a great role in all three phases of immunoediting—(1) Elimination, (2) Equilibrium, and (3) Escape (the three Es) [23,24].

HLA-G exerts its immunosuppressive activity by various mechanisms, e.g., (1) through binding to inhibitory receptors on immune cells. These inhibitory receptors like immunoglobulin-like transcript (ILT) 4, are negative regulators of immune response not only in allograft rejection, autoimmunity, and infectious diseases but also in tumor development [25]. To date, several inhibitory receptors for HLA-G have been identified, such as CD85j/immunoglobulin-like transcript 2 (ILT2), CD85d/ILT4, and CD158d/killer cell immunoglobulin-like receptor 2DL4 (KIR2DL4). Moreover, CD8 and CD160 have also been reported to strongly bind to HLA-G [26]. HLA-G even up-regulates ILT2, ILT3, ILT4, KIR2DL4, and LILRB1 inhibitory receptors in antigen-presenting cells, NK cells, and T cells [25,27]. Through binding of both membrane-bound or soluble HLA-G to inhibitory receptors on immune cells, HLA-G directly inhibits the functions of these effector cells leading to immune suppression, e.g., NK cells [25,28,29] cytotoxic T-lymphocytes (CTL) [30], T and B cells [31,32]—monocytes, macrophages, neutrophils, myeloid-derived stem cells (MDSC), CD8^+^PD-1^−^ILT2^+^ intra-tumoral T cells [33], and dendritic cells (DC) [34,35].

Cai et al. (2019) in their study showed the co-expression of ILT4 and HLA-G in tissues of human primary colorectal cancer (CRC). They revealed that the co-expression of ILT4 and HLA-G in tissues of human primary CRC and their mutual interaction promotes the progression of human colorectal cancer. The association of HLA-G with older age, advanced stage, regional lymph node involvement, and poor overall survival time was observed [36].

The binding of sHLA-G to CD8 molecules of cytotoxic T cells and NK cells induces their apoptosis [37,38]. Ajith et al. (2019) demonstrated a novel mechanism by which HLA-G dimer inhibits activation and cytotoxic capabilities of human CD8^+^ T cells. This mechanism implicated the down-regulation of granzyme B expression and the essential involvement of the inhibitory receptor LILRB1 [39]. The same mechanism might have a negative anti-tumor effect.

As an indirect effect, HLA-G inhibits the proliferation of allo-specific T cells, maturation, and function of B cells and may lead to the generation of HLA-G expressing tolerogenic DC-10 [32,40,41]. HLA-G^+^ antigen-presenting cells can induce immunosuppressive CD4^+^ T cells and, in the case of DC-10, HLA-G mediates the generation of type 1 regulatory T cells [41,42].

(2) The second mechanism of immunosuppressive activity of HLA-G is done by transferring this immunosuppressive molecule from tumor or immune cells to other immune cells through trogocytosis, exosomes, and tunneling nanotubes. This way, HLA-G^acq+^ CD4^+^ T cells, HLA-G^acq+^ CD8^+^ T cells, HLA-G^acq+^ NK cells, and HLA-G^acq+^ CD14^+^ monocytes with immunosuppressive and regulatory activity are formed, further amplifying the tolerogenic effects of HLA-G in tumor immune escape which leads to immune evasion, therapy resistance, disease progression and poor clinical outcomes [43]. Induction of these HLA-G^+^ tolerogenic cells, including DC-10, leads to long-lasting immune-regulatory activities [12,22].

Due to its immunosuppressive activity, the negative role of HLA-G has been found to be related to the development of the tumor process. HLA-G overexpression has been associated with the development of several solid tumors and contributed to their immune evasion [44]. Its expression has also been correlated with poor clinical outcomes in cancer patients [44,45].

Many studies have claimed HLA-G as a new immune checkpoint in cancer [46]. Recent medical investigations suggest that HLA-G can be used as a biomarker in the diagnosis, treatment, and prognosis of different neoplasms. However, the potential diagnostic value of sHLA-G with other tumor markers in gliomas has not been explored yet.

As the expression of HLA-G is finely tuned by genetic variations (polymorphisms) in the non-coding region of the locus, besides plasma levels of soluble HLA-G in glioma patients, we also investigated the gene polymorphism of this molecule in glioma patients. An important polymorphism is the presence of the 14 nt sequence (insert, rs16375) in the 3′ non-transcribed region of the HLA-G gene, as it was found to negatively affect the stability of mRNA and hence total serum soluble HLA-G (sHLA-G) levels [47,48,49]. We were also interested in whether and how the plasma levels of sHLA-G are related to the methylation of the MGMT promoter, which affects the efficacy of treating glioma patients [50]. We also evaluated a possible correlation of sHLA-G with the level of immunoregulatory and anti-inflammatory cytokine IL-10, pro-inflammatory cytokine IL-6, and their ratio (IL-10/IL-6), and the association of sHLA-G with the survival of glioma patients. Indeed, in addition to good cell-mediated immunity, it is precisely inflammation that plays an important role in the pathogenesis of gliomas [51].

## 2. Subjects and Methods

### 2.1. Study Groups

A total of 59 patients (25 women and 34 men) with gliomas were enrolled in our study. Out of them, 49 had primary gliomas and 10 recurrent gliomas. The mean age of the patients at the time of diagnosis was 53.36 ± 15.17 years. A total of 19 patients had glioma grade II, 11 grade III, and 29 glioma grade IV (Table 1). The control group compromises 159 healthy subjects without cancer diagnosis (80 women and 79 men with a mean age of 40.91 ± 11.93 years).

### 2.2. Procedures and Sample Processing

A total of 10 mL of blood was collected from glioma patients in a tube with EDTA (ethylenediaminetetraacetic acid) and 5 mL of blood in a tube without anticoagulant on the morning of surgery at the Department of Neurosurgery of the Faculty of Medicine, Comenius University, and University Hospital in Bratislava. Approximately 1.5 mL of blood was collected from an EDTA tube and immediately transferred to the Laboratory of Immunology, Medirex, Ltd. The remaining samples were immediately centrifuged in our laboratory and the plasma (from the EDTA tube) or serum from the tube without anticoagulant was withdrawn, aliquoted into microtubes, and stored in a deep-freezer box at −80 °C until examination. DNA was isolated from the remaining blood (after the withdrawal of the plasma) by the salting out procedure [52] and stored at −20 °C. Later, this DNA was used to examine the HLA-G 14bp insertion/deletion polymorphism in the 3′ UTR (rs16375). The study was conducted in accordance with the Declaration of Helsinki, approved by the Ethical Committee of the Faculty of Medicine, Comenius University, and University Hospital in Bratislava (project identification code: 17/2015). Each patient received written informed consent. The patients came from the Department of Neurosurgery of the Faculty of Medicine, Comenius University, and the University Hospital in Bratislava. The histopathological diagnosis and grade of malignancy were investigated at Cytopathos, Ltd. and Alpha Medical, Ltd. in Bratislava.

The HLA-G 14bp ins/del polymorphism was investigated by polymerase chain reaction (PCR) as described by Hviid et al. (2002) [48]. Briefly, DNA was amplified by forward primer 5′GTGATGGGCTGTTTAAAGTGTCACC-3′ and reverse primer 5′GGAAGGAATGCAGTTCAGCATGA-3′ using a PCR cycler (Biometra, Jena, Germany). The reaction mixture with a total volume of 25 μL contained 50 ng of template DNA, 0.2 mM of each dNTP (Thermo Fisher Scientific, Waltham, MA, USA), 1 unit of Taq DNA polymerase (Thermo Fisher Scientific), 1.5 mmol MgCl2 (Thermo Fisher Scientific), and 10 pmol of each specific primer. PCR conditions were 95 °C for 3 min, followed by 30 cycles (denaturation at 95 °C for 1 min, annealing at 64 °C for 1 min, and elongation at 72 °C for 1 min), and final elongation at 72 °C for 10 min. The PCR products were run in 3% agarose gel for 20 min and then visualized under UV light. Fragment size was confirmed using the 100 bp DNA ladder (SBS). PCR fragments of 224 bp (14bp insertion) and PCR fragments of 210 bp (14bp deletion) were identified.

The plasma level of soluble HLA-G (sHLA-G) was determined by sandwich ELISA according to the manufacturer’s recommended procedure (human sHLA-G ELISA kit; Exbio, BioVendor, Brno, Czech Republic). The soluble HLA-G levels were analyzed in the plasma of 59 patients with brain gliomas and 43 healthy controls. The selection criteria for the control group were plasma availability of healthy subjects that fulfilled the criteria of the absence of brain tumors or other malignancies, the absence of autoimmune diseases, allergies (also in the family), and control subjects had no signs of acute inflammation.

The methylation analysis of the MGMT promoter was performed by a pathologist in the tumor tissue by bisulfite conversion-specific and methylation-specific PCR so that it cannot be examined in healthy controls [53]. It was investigated by pathologists in all G IV and in some G III glioma patients; it is not always investigated at lower grades, as this is especially important for the treatment of G. IV glioma patients.

Concentrations of anti-inflammatory cytokine IL-10 (human IL-10 Elisa kit; Wuhan Fine Biotech Co., Ltd., Wuhan, China) and pro-inflammatory cytokine IL-6 in plasma of glioma patients (human IL-6 Elisa kit; Wuhan Fine Biotech Co., Ltd., Wuhan, China) were also determined by the sandwich ELISA method. IL-6 and IL-10 levels were examined in the plasma of 32 patients with gliomas that were still available. Therefore, correlations of sHLA-G levels with IL-6 and IL-10 levels were also determined in 32 patients, those who had all three parameters −levels of s HLA-G, IL-6, and IL-10.

The survival time was calculated from the time of diagnosis until April 2019 or the time of death. Patients were monitored from 1 December 2015 to 30 April 2019.

### 2.3. Statistical Analysis

Statistical significance of differences in allele and genotype frequencies between two studied groups (gliomas vs. controls) was evaluated by the standard chi-square test using the InStat statistical software (GraphPad Software, Inc., version 3.10, San Diego, CA, USA). The odds ratios (OR) and 95% confidence intervals (95% CI) were calculated as well. Finally, multivariate logistic regression analyses adjusted for sex and age as possible influencing factors were performed by the SNP stats web software available at https://www.snpstats.net/start.htm (accessed on 20 February 2018) [54].

For statistical analysis, we used the programs InStat and SAS Enterprise Guide 6.1. We used Student’s *t*-test, the Mann–Whitney test, Cox proportional hazard analysis, Kaplan–Meier survival analysis, and Log Rank test. The Spearman correlation has also been tested. The results were expressed as the median and interquartile range (IQR), mean ± standard deviation (SD), and hazard ratio (HR). A *p*-value < 0.05 was considered to indicate the statistical significance.

## 3. Results

### 3.1. Comparison of HLA-G 14bp Ins/Del Allele and Genotype Frequencies between Glioma Patients and Control Group of Healthy Subjects

In a group of 59 patients with brain gliomas and 159 controls, the presence of 14 nt insert (14 ins) in the 3′UTR region of HLA-G by PCR was analyzed. The differences in the frequencies of the HLA-G alleles and genotypes carrying the 14 nt insert between the examined groups were calculated using the chi-square test. The group of glioma patients comprised a significantly higher proportion of individuals carrying the 14 nt insert in both homozygous and heterozygous forms (14 ins/ins and 14 del/ins) compared to the control group of healthy subjects (79.66% vs. 65.41%; *p* = 0.03; Table 2). After adjustment for age and sex, no statistically significant association of HLA-G 14bp ins/del variants with gliomas was found.

### 3.2. Comparison of Plasma Levels of sHLA-G in Glioma Patients and Healthy Controls

Next, we analyzed the level of immunosuppressive molecule sHLA-G in a group of patients with gliomas and healthy controls. We found significantly higher plasma levels of sHLA-G in glioma patients compared to the healthy control population (*p* = 0.048; Table 3).

We were also interested in whether sHLA-G levels differ in patients with gliomas at different stages of the disease. However, no statistically significant differences in sHLA-G levels among different gliomas grades were found (Table 4).

Comparing the plasma levels of sHLA-G in patients with primary and relapmed gliomas, our results revealed higher levels of sHLA-G in relapsed glioma patients (Median ± IQR: 27.57 U/mL ± 35.79) than in primary gliomas (median ± IQR: 46.66 U/mL ± 54.66), however, the difference was not statistically significant (*p* = 0.071).

### 3.3. The Effect of HLA-G 14bp Ins/Del Polymorphism on sHLA-G Plasma Levels in Glioma Patients

In the group of patients with gliomas, we compared the level of sHLA-G with HLA-G 14bp ins/del variants to determine if the investigated polymorphism affects the plasma sHLA-G level. The level of sHLA-G between individuals with different variants of the HLA-G bp 14 ins/del genotypes was not statistically different (*p* = 0.395–0.957; Table 5).

### 3.4. Analysis of the Association of Plasma Levels of sHLA-G with a O^6^-Methylguanine-DNA Methyl-Transferase (MGMT) Promoter Methylation Status in Glioma Patients

Comparing plasma levels of sHLA-G in a group of patients with methylated and unmethylated MGMT promoters, patients with methylated MGMT promoters were shown to have lower plasma levels of the immunosuppressive molecule sHLA-G than those with unmethylated MGMT promoters. This suggests that patients with methylated MGMT promoters who respond better to treatment have significantly lower plasma concentrations of sHLA-G (mean: 29.51 U/mL vs. 54.30 U/mL; *p* = 0.03; Table 6).

### 3.5. Association of Plasma Levels of Immunosuppressive Molecule sHLA-G with the Levels of Pro-Inflammatory Cytokine IL-6, Anti-Inflammatory Cytokine IL-10, and IL-10/IL-6 Ratio in Glioma Patients

Since inflammation plays an important role in the pathogenesis of gliomas, we wondered whether the level of sHLA-G in plasma affects the concentrations of the selected pro- and anti-inflammatory cytokines. The results showed that the levels of sHLA-G negatively highly significantly correlated with the pro-inflammatory cytokine IL-6 concentration (Table 7, *p* = 0.0004). However, we did not find a correlation with the level of the anti-inflammatory cytokine IL-10 but found a positive correlation with the ratio of IL-10/IL-6 levels in the plasma of glioma patients (*p* = 0.046; Table 7).

### 3.6. Association of Plasma Levels of sHLA-G with Survival Time in Grade II and IV Glioma Patients

When we performed Cox hazard proportional analyses, sHLA-G was proven to have influence on overall survival time in grade II (HR = 1.023; *p* = 0.0088) and IV (HR = 1.004; *p* = 0.0399) glioma patients (Table 8). Survival time is reduced with plasma sHLA-G increase. In grade III glioma patients, this association was not observed.

### 3.7. Correlation between Plasma Level of sHLA with Overall Survival in Whole Group of Glioma Patients and in the Subgroup of Glioblastoma Patients

The plasma level of sHLA-G negatively correlated with overall survival in the whole group of glioma patients (Spearman r = −0.25, *p* = 0.05, 95% CI: −0.4825–0.0085). Glioblastoma patients (G IV) who survived more than one year after diagnosis had significantly lower plasma values of sHLA-G than patients who survived less than one year (median 21.5 U/mL vs. 46.74 U/mL, *p* = 0.02) (Table 9). We also observed a significant difference in overall survival of G IV patients when we compared patients with sHLA-G above and below cut off of sHLA-G 40 U/mL. Patients with sHLA-G levels below 40 U/mL survived significantly longer than patients with sHLA-G above 40 U/mL (*p* = 0.038, Figure 1). In G II the difference did not reach the statistical significance (*p* = 0.06) and in G III was not significant at all, probably due to the low number of patients.

When we compared the mean survival time (±SD) in patients with sHLA-G below and above 40 U/mL in grade IV of gliomas, we found that in patients (*n* = 17) with sHLA-G levels below 40 U/mL the mean survival time (±SD) was 12 ± 9.82; 95% CI: 6.95–17.05, and in patients with sHLA-G levels above 40 U/mL it was (*n* = 12): 6.5 ± 3.97; 95% CI = 3.98–9.02.

We also determined the progression-free survival (PFS) of grade IV glioma patients according to the plasma level of sHLA-G. Comparing the 30 month PFS in patients with sHLA-G below 40 U/mL (*n* = 17; mean and standard error/SE/: 0.1250 (0.0827)) with patients with sHLA-G above 40 U/mL (*n* = 12; mean and standard error/SE/: 0.0833 (0.0798)) we did not find a statistical significant difference (test: Log rank *p* = 0.5104) (Figure 2).

## 4. Discussion

From an immunological point of view, the development and progression of tumors is influenced by two main immune factors: decreased Th1 immunity and increased inflammatory process or presence of at least chronic low-grade inflammation. Many studies have claimed HLA-G as a new immune checkpoint in cancer [46]. Emerging evidence indicates that tissue expression of HLA-G, as well as HLA-G-expressing regulatory cells, and the level of soluble HLA-G may play an important role in dictating the outcome of the anti-tumor immune response [55].

HLA-G expression in tumor lesions was first demonstrated in melanoma [56] and later, its expression has been correlated with poor clinical outcomes in various cancer patients [44,45,57,58,59,60]. In recent years there have been some reports that point to its importance in patients with gliomas [45].

In our previous study analyzing HLA-G 5′URR, we found a genetic association of haploblock consisting of −762 T, −716 G, −689 G, −666 T, and −633 A allele followed by −486 C and −201 A alleles with susceptibility to developing gliomas for the first time. In the grade IV glioma patients, we also observed that haploblock carriers of −762 CT, −716 TG, −689 AG, −666 GT, −633 GA, −486 AC, −477 GC, −201 GA followed by −369 AC carriers tend to have lower age at onset as compared to other genotype carriers. However, no correlation of HLA-G 5′URR variants with sHLA-G plasma level was found [61]. There is only one study by Magalhaes et al. (2021) analyzing the association of the 14bp ins/del polymorphisms of the HLA-G 3′ UTR and its relationship with plasma sHLA-G level in glioma patients published in April 2021 [62]. As it is the first study concerning HLA-G 14bp ins/del polymorphism in gliomas, our study is the second one.

We analyzed the presence of 14 nt insert in the 3′UTR region of HLA-G and found a significantly higher proportion of glioma patients carrying the 14 nt insert in both homozygous and heterozygous states (14 ins/ins and 14 del/ins) compared to the control group of healthy subjects (*p* = 0.03). Lau et al. (2011) reported no association of 14bp ins/del polymorphism with a risk of childhood neuroblastoma and their analyses did not detect an association between common HLA-G polymorphisms and clinical outcomes in patients treated for neuroblastoma [63]. The prevalence of the HLA-G genotype carrying the 14 nt insert was also found in patients with other solid tumors [64,65,66,67]. However, there is also some opposite evidence that the presence of the HLA-G 14 nt insert is associated with a reduced risk of developing malignancies [68].

Both Magalhães’ (studied gliomas grade IV) and our study revealed higher levels of soluble HLA-G—we, in the group of all glioma patients (with *p*-value = 0.048), and Magalhães et al. [62] in gliomas of grade IV—the worst grade of gliomas. As we can see (Table 4), the plasma levels of sHLA-G increased in patients with glioma gradings, probably with an increased number of gliomas of grade IV, the *p*-value would be more significant also in our study. Glioma patients in our study compared to healthy controls had also higher plasma levels of sHLA-G (*p* = 0.009), which is consistent with the results of other studies [69].

To determine if the investigated polymorphism affects the plasma level of sHLA-G in glioma patients, we compared the level of sHLA-G in patients with different variants of the HLA-G 14 ins/del genotypes; however, the differences were not statistically significant. It might be explained by the fact that the levels of sHLA-G in glioma patients are influenced not only by HLA-G variants, but also by environmental factors, such as treatment, hormones, stress, and hypoxia [70,71]. However, Magalhaes et al. (2020) found an association of the heterozygous 14bp ins/del and +3142 C/G genotypes of the HLA-G 3′ UTR with higher HLA-G plasma levels in grade IV glioma patients when compared with controls [62].

Comparing plasma levels of sHLA-G in our group of patients with methylated and unmethylated MGMT (methylguanidine methyltransferase) promoters, patients with methylated MGMT promoters were shown to have lower plasma levels of the immunosuppressive molecule sHLA-G than those with unmethylated MGMT promoter. This suggests that patients with methylated MGMT promoters who respond better to treatment have significantly lower plasma concentrations of sHLA-G (*p* = 0.03). The explanation for this is the knowledge that patients with a methylated MGMT promoter are known to respond better to treatment with alkylating cytostatics, e.g., temozolomide. Temozolomide alkylates DNA bases and damages tumor cells. If this enzyme (MGMT) is functional in the patient, it corrects what temozolomide kills and thus worsens the patient’s prognosis. Patients with a methylated MGMT promoter have a less functional MGMT enzyme and respond better to treatment with particular temozolomide [72]. We did not find any study concerning the association of the level of sHLA-G with the methylation status of MGMT.

Further, we proved the influence of plasma sHLA-G on overall survival time in grade II (*p* = 0.0088) and grade IV (*p* = 0.0399) glioma patients. Survival time is reduced with plasma sHLA-G increase. We also analyzed the correlation between plasma level of sHLA-G with overall survival in the whole group of glioma patients and in the subgroup of glioblastoma (G IV) patients. We found that the plasma level of sHLA-G negatively correlated with overall survival in the whole group of glioma patients (*p* = 0.05). Glioblastoma patients who survived more than one year after diagnosis had significantly lower plasma values of sHLA-G than patients who survived less than one year (*p* = 0.02). We also observed a significant difference in overall survival of G IV patients when we compared patients with sHLA-G above and below the cut-off of sHLA-G 40 U/mL (*p* = 0.038). Patients with an sHLA-G level below 40 U/mL survived significantly longer than patients with sHLA-G above 40 U/mL.

There exists consistent evidence in the literature that plasma levels of sHLA-G are higher in cancer patients than in healthy controls. This was proven in breast cancer, gastrointestinal tumors, lung, and urogenital cancer [73,74,75]. Additionally, Kirana et al. (2017) and others found an association between higher sHLA-G levels and worse prognosis in colorectal cancer [76,77].

In our study, we observed higher plasma levels of sHLA-G in patients with gliomas than in healthy controls. We think that soluble HLA-G could be released from tumors to help them escape from immune surveillance of the body. We suppose that the increased level of immunosuppressive sHLA-G in the peripheral blood inhibits the anti-tumor immunity, helps tumor growth, promotes faster progression, and shorter overall survival. This hypothesis is supported by our finding that patients with GBM who survived less than one year had significantly higher values of sHLA-G.

Since inflammation plays an important role in the pathogenesis of gliomas, we wondered whether the level of sHLA-G in plasma affects the concentrations of the selected pro- and anti-inflammatory cytokines. The results showed that the levels of sHLA-G negatively highly significantly correlated with the concentration of pro-inflammatory cytokine IL-6 (*p* = 0.0004), thus likely contributing to suppressing its production. However, we did not find a correlation with the level of the anti-inflammatory cytokine IL-10 (sHLA-G does not affect its formation) but found a positive correlation with the ratio of IL-10/IL-6 levels in the plasma of glioma patients (Table 9; *p* = 0.046). We found a study in which authors showed that IL-10 increases the expression of HLA-G [78].

We realize that a larger patient and control group would have greater statistical power, however, the number of study subjects was limited by their availability. We assume that our present-day results as useful for future investigations on this topic. HLA-G plays a great role in the immunopathogenesis of cancer, and probably also gliomas, and the plasma level of sHLA-G might have an impact on the survival of glioma patients.

## 5. Conclusions

A higher proportion of glioma patients had the 14 nt insert at the HLA-G 3′UTR in both homozygous and heterozygous states compared to the control group. Glioma patients also had higher plasma levels of sHLA-G. The level of this immunosuppressive and anti-inflammatory molecule was lower in patients with methylated MGMT promoters than those with unmethylated MGMT promoters. This finding has not been described so far. We proved the influence of plasma sHLA-G on overall survival time in grade II and IV glioma patients. Survival time is reduced when plasma sHLA-G increases. The plasma level of sHLA-G is negatively correlated with overall survival in the whole group of all glioma patients. Glioblastoma patients who survived more than one year after diagnosis had significantly lower plasma values of sHLA-G than patients who survived less than one year. The overall survival of G IV patients with the level of sHLA-G below 40 U/mL was significantly longer than of patients with sHLA-G above 40 U/mL. We are aware of a limited number of glioma patients and recognize that there are many factors that can affect their survival, in our study, we focused only on HLA-G. The levels of sHLA-G are also highly negatively correlated with the concentration of pro-inflammatory cytokine IL-6 and positively with the IL-10/IL-6 ratio in plasma of glioma patients.

## Figures and Tables

**Figure 1 diagnostics-12-01099-f001:**
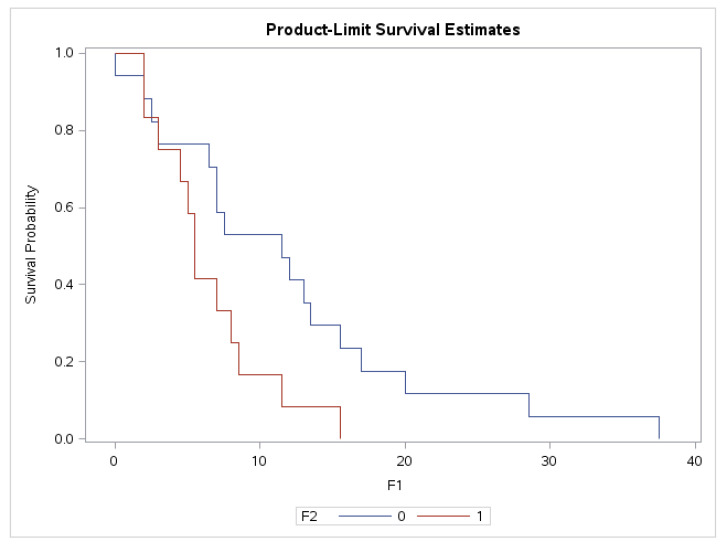
Kaplan–Meier survival curves of grade IV according to sHLA-G. 0—patients with sHLA-G < 40 (*n* = 17), 1—patients with sHLA-G > 40 (*n* = 12), X-axis—survival time in months, Y-axis—survival probability (test: Log rank *p* = 0.038).

**Figure 2 diagnostics-12-01099-f002:**
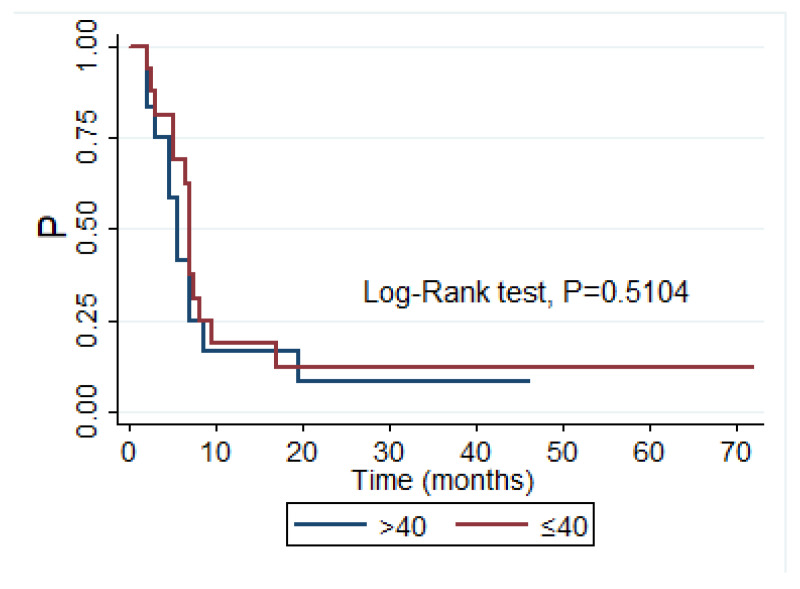
Kaplan–Meier 30 months progression survival curves of grade IV glioma patients according to sHLA-G, patients with sHLA-G < 40 (*n* = 17), patients with sHLA-G > 40 (*n* = 12), X-axis—progression-free survival time in months, Y-axis—survival probability in months without progression (test: Log rank *p* = 0.5104).

**Table 1 diagnostics-12-01099-t001:** Characteristics of the study group of glioma patients.

Parameter (Mean ± SD)	Brain Gliomas *N* = 59
Age at diagnosis	53.36
Sex (women/men)	25/34
Grade of gliomas	
Grade II	19
Grade III	11
Grade IV	29
Primary diagnosis of glioma	49
Relapse	10

*N*—number of patients; SD—standard deviation.

**Table 2 diagnostics-12-01099-t002:** Allele and genotype frequencies of HLA-G 14bp ins/del polymorphism in glioma patients and healthy controls.

Allele/Genotype	Brain Gliomas (*N* = 59)	Controls (*N* = 159)	Univariate Analysis		Multivariate Analysis	
			*p*	OR (95% CI)	*p*	OR (95% CI)
−14	58 (49.15%)	191 (60.06%)				
+14	60 (50.85%)	127 (39.94%)	0.05	1.56 (1.02–2.38)	-	-
−14/−14	12 (20.34%)	55 (34.59%)		1.00		1.00
−14/+14	34 (57.63%)	81 (50.94%)	0.09	1.92 (0.92–4.04)	0.35	1.64 (0.72–3.72)
+14/+14	13 (22.03%)	23 (14.47%)		2.59 (1.03–6.52)		2.00 (0.71–5.65)
−14/−14	12 (20.34%)	55 (34.59%)		1.00		1.00
−14/+14 and +14/+14	47 (79.66%)	104 (65.41%)	**0.03**	2.07 (1.02–4.23)	0.17	1.72 (0.78–3.78)
−14/−14 and −14/+14	46 (77.97%)	136 (85.53%)		1.00		1.00
+14/+14	13 (22.03%)	23 (14.47%)	0.19	1.67 (0.78–3.57)	0.43	1.43 (0.60–3.40)
−14/−14 and +14/+14	25 (42.37%)	78 (49.06%)		1.00		1.00
−14/+14	34 (57.63%)	81 (50.94%)	0.38	1.31 (0.72–2.39)	0.54	1.24 (0.63–2.44)

Allele and genotype frequencies are presented as absolute numbers with percentages in parentheses. Univariate analysis is based on χ^2^ tests. Multivariate analysis is adjusted by sex and age. OR—odds ratio; CI—confidence interval, *n*—number, +14bp—insertion of 14bp sequence, −14bp—deletion of 14bp sequence.

**Table 3 diagnostics-12-01099-t003:** The plasma level of sHLA-G in glioma patients and healthy controls.

	Patients (*N* = 59)	Healthy Controls (*N* = 43)	*p* (Student’s *t*-Test)
sHLA-G (U/mL) (Mean ± SD)	42.17 ± 38.50	23.06 ± 9.53	**0.048**

sHLA-G—soluble HLA-G, SD—standard deviation, *p* ≤ 0.05 is statistically significant.

**Table 4 diagnostics-12-01099-t004:** Comparison of plasma levels of sHLA-G in different stages of glioma in primary diagnosis (*n* = 49).

	Grade 2 *N* = 13	Grade 3 *N* = 9	Grade 4 *N* = 27	*p* (2 vs. 3)	*p* (2 vs. 4)	*p* (3 vs. 4)
sHLA-G (U/mL) (Mean ± SD)	39.19 ± 40.96	32.05 ± 13.69	43.23 ± 39.42	0.45	0.22	0.70

sHLA-G—soluble HLA-G, SD—standard deviation, *p* ≤ 0.05 is statistically significant.

**Table 5 diagnostics-12-01099-t005:** Association between HLA-G 14bp ins/del polymorphism and the level of sHLA-G in glioma patients (*n* = 59).

Allele/Genotype	sHLA-G (U/mL) Mean ± SD	*p*
A. −14 (*n* = 58)	44.586 ± 38.34	
B. +14 (*n* = 60)	37.942 ± 29.81	0.499 (A vs. B)
C. −14/−14 (*n* = 12)	45.76 ± 36.17	0.395 (C vs. D)
D. −14/+14 (*n* = 34)	41.06 ± 35.77	0.468 (D vs. E)
E. +14/+14 (*n* = 13)	48.56 ± 39.06	0.957 (C vs. E)

*N*—number of patients; sHLA-G—soluble HLA-G, SD—standard deviation; *p* ≤ 0.05 is statistically significant.

**Table 6 diagnostics-12-01099-t006:** Plasma levels of sHLA-G in glioma patients with different MGMT promoter methylation statuses.

MGMT Promoter	Patients (*N* = 32)	sHLA-G (U/mL) (Mean ± SD)	*p*
methylated	17	29.51 ± 23.50	**0.03**
unmethylated	15	54.30 ± 43.12	

*N*—number of patients; sHLA-G—soluble HLA-G, SD—standard deviation; *p* ≤ 0.05 is statistically significant.

**Table 7 diagnostics-12-01099-t007:** Correlation of plasma levels of sHLA-G with IL-6, IL-10 levels, and IL-10/IL-6 in glioma patients.

Patients (*N* = 32)	Correlation of sHLA-G with IL-6	Correlation of sHLA-G with IL-10	Correlation of sHLA-G with IL-10/IL-6 Ratio
Spearman r	−0.584	0.208	0.622
95% CI	−0.779–(−0.286)	−0.527–0.163	−0.0034–0.633
** *p* **	**0.0004**	**0.26**	**0.046**

*N*—number of patients, sHLA-G—soluble HLA-G; CI—confidence interval, *p* ≤ 0.05 is statistically significant.

**Table 8 diagnostics-12-01099-t008:** Cox proportional hazard analysis of sHLA-G and survival time in grade II and IV glioma patients.

Parameter		Parameter Estimate	Standard Error	Chi-Square	*p*	Hazard Ratio
sHLA-G in G. II (*n* = 19)	Survival time (months)	0.02234	0.00852	6.8696	0.0088	1.023
sHLA-G in G. IV (*n* = 29)	Survival time (months)	0.00435	0.00212	4.2222	0.0399	1.004

**Table 9 diagnostics-12-01099-t009:** Comparison of sHLA-G in patients with G. IV who survived more than one year and less than one year (nonparametric test).

sHLA-G (U/mL)	Pts. Surviving < 1 Year *N* = 21	Pts. Surviving > 1 Year *N* = 8
Median	46.74	21.50
IQR	74.80	12.60
95% CI	30.02–119.10	14.59–30.17
*p*	**0.02**	

*N*—number, G—grade, Pts.—patients, CI—confidence interval, *p* < 0.05 is statistically significant.

## Data Availability

The data presented in this study are available on request from the corresponding author.

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
