# Peer review of "HLA-G 14bp Ins/Del Polymorphism, Plasma Level of Soluble HLA-G, and Association with IL-6/IL-10 Ratio and Survival of Glioma Patients"

_diagnostics, 2022, doi:10.3390/diagnostics12051099_

Round 1
Reviewer 1 Report
No further comments
Author Response
Response to reviewer 1
HLA-G 14bp ins/del Polymorphism, Plasma Level of Soluble HLA-G and Association with IL-6/IL-10 Ratio and Survival of Glioma Patients
Dear reviewer, thank you very much for reviewing our article. The native speaker has corrected our article.
This reviewer did not have any comments.
Thank you very much. Best regards
Bucova Maria

Reviewer 2 Report
The title reflects the subject of the study. In this article, Bucova and others revealed the relationship between sHLA-G and the malignant degree of gliomas, and the levels of sHLA-G negatively correlated with the concentration of pro-inflammatory cytokine IL-6 and positively with IL-10/IL-6 ratio in plasma of glioma patients.
However, the following issue should be addressed:
(1)The authors measured plasma level of soluble HLA-G by sandwich ELISA, and was it the same as the direct detection of HLA-G expression in tumors?
(2)Table4 showed the comparison of plasma levels of sHLA-G in different stages of glioma in primary diagnosis, I suggest that the author should put the information of recurrent gliomas into the table, or make a comparison between newly diagnosed gliomas and recurrent gliomas.
(3)In the Table6, authors compared the plasma levels of sHLA-G in glioma patients with different MGMT promoter methlyation status. In the Table7, authors described the correlation of plasma levels of sHLA-G with IL-6, IL-10 levels and IL-10/IL-6 in glioma patients. Why the case was only 32? Please add the exclusion reason to the manuscript.
(4)Figure1 and Table10 expressed the same content, and one of them is supposed to be deleted. It could be better to add the progression free survival of glioma patients to compare it.
(5)Inflammation plays an important role in the pathogenesis of gliomas, authors could supplement the relationship between HLA-G and IL-6 or IL-10 into the introduction or discussion part to enrich the content.
Finally, because of a low number of patients, it is questionable to draw any conclusion. Present works did not receive a high enough priority, it is suggested to supplement the sample size to enrich the content of this article.

Author Response
Response to reviewer 2
HLA-G 14bp ins/del Polymorphism, Plasma Level of Soluble HLA-G and Association with IL-6/IL-10 Ratio and Survival of Glioma Patients
The title reflects the subject of the study. In this article, Bucova and others revealed the relationship between sHLA-G and the malignant degree of gliomas, and the levels of sHLA-G negatively correlated with the concentration of pro-inflammatory cytokine IL-6 and positively with IL-10/IL-6 ratio in plasma of glioma patients. However, the following issue should be addressed:
(1)The authors measured plasma level of soluble HLA-G by sandwich ELISA, and was it the same as the direct detection of HLA-G expression in tumors?
Answer:
- The plasma level of sHLA-G was detected by the Elisa test, we did not detect the expression of HLA-G directly in gliomas. The level of sHLA-G is the response from the periphery and elevated levels of sHLA-G reflect the immune suppression in the periphery, which is in concordance with worse survival of these patients.
(2)Table4 showed the comparison of plasma levels of sHLA-G in different stages of glioma in primary diagnosis, I suggest that the author should put the information of recurrent gliomas into the table, or make a comparison between newly diagnosed gliomas and recurrent gliomas.
Answer:
- Thank you very much for your comment, we add comparative results between primary and recurrent gliomas to the text after Table 4. It is marked in blue: Comparing the plasma levels of sHLA-G in patients with primary and relapsed gliomas our results revealed higher levels of sHLA-G in relapsed glioma patients (Median ± IQR: 27.57 U/mL ±79) than in primary gliomas (Median ± IQR: 46.66 U/mL ± 54.66), however, the difference was not statistically significant (P=0.071).
- Here is the table with our results comparing levels of sHLA-G in patients with primary and recurrent gliomas.
Table … Comparison of sHLA-G in patients with primary diagnosis and relapsed glioma
|
|
Primary gliomas (N=49) |
Relapsed gliomas (N=10) |
P |
|
Median s HLA-G (U/mL) |
27.57 |
46.66 |
0.071 |
|
IQR |
35.79 |
54.66 |
|
(3)In the Table6, authors compared the plasma levels of sHLA-G in glioma patients with different MGMT promoter methlyation status. In the Table7, authors described the correlation of plasma levels of sHLA-G with IL-6, IL-10 levels and IL-10/IL-6 in glioma patients. Why the case was only 32? Please add the exclusion reason to the manuscript.
Answer:
- The methylation analysis of the MGMT promoter was investigated by pathologists in all G IV and in some G III glioma patients. It is not always investigated at lower grades, as this is especially important for the treatment of G. IV glioma patients. This text was added to the section „Subjects and Methods“ and is marked in blue.
- IL-6 and IL-10 levels were examined in the plasma of 32 patients with gliomas that were still available. Plasmas from the rest of the patients were not available, they were used before to determine other parameters (BDNF, GDNF, VEGF, sTREM-1, fractalkine, IL-15, ....). We add this bold typed sentence in the section “Subjects and Methods” - it is marked in blue.
(4)Figure1 and Table10 expressed the same content, and one of them is supposed to be deleted. It could be better to add the progression free survival of glioma patients to compare it.
- We deleted Table 10, instead of it, we add some information to the text. The text follows the Figure 1 and is marked in blue: „When we compared the mean survival time (± SD) in patients with sHLA-G below and above 40 U/mL in grade IV of gliomas, we found that in patients (N=17) with the level of sHLA-G below 40 U/mL was the mean survival time (± SD) 12 ± 9.82; 95% CI: 95 – 17.05, and in patients with the level of sHLA-G above 40 U/mL (N=12): 6.5 ± 3.97; 95% CI: 3.98 - 9.02.”
- The Figure 2 concerning progression free survival in grade IV of glioma patients with sHLA-G below and above 40 U/mL has been added to the article.
- We add also a text concerning these results before Figure 2 – marked in blue: We determined also the progression free survival (PFS) of grade IV glioma patients according to the plasma level of sHLA-G. Comparing the 30 months PFS in patients with sHLA-G below 40 U/mL (N = 17) with patients with sHLA-G above 40 U/mL (N = 12) we did not find a statistical significant difference (Test: Log rank P = 0.5104) (Figure 2).
(5)Inflammation plays an important role in the pathogenesis of gliomas, authors could supplement the relationship between HLA-G and IL-6 or IL-10 into the introduction or discussion part to enrich the content.
- The penultimate section of the "Discussion" section contains information about inflammation and these cytokines, which are marked in blue: Since inflammation plays an important role in the pathogenesis of gliomas, we wondered whether the level of sHLA-G in plasma affects the concentrations of the selected pro- and anti-inflammatory cytokine. The results showed that the levels of sHLA-G negatively highly significantly correlated with the concentration of pro-inflammatory cytokine IL-6 (P = 0.0004), thus likely contributing to suppressing its production. However, we did not find a correlation with the level of the anti-inflammatory cytokine IL-10 (sHLA-G) does not affect its formation), but found a positive correlation with the ratio of IL-10 / IL-6 levels in the plasma of glioma patients (Table 9; P = 0.046). We found a study, in which authors showed, that IL-10 increases the expression of HLA-G [78].
- There is also some information at the end of “Introduction”: We also evaluated a possible correlation of sHLA-G with the level of immunoregulatory and anti-inflammatory cytokine IL-10, pro-inflammatory cytokine IL-6, and their ratio (IL-10 / IL-6), and the association of sHLA-G with the survival of glioma patients. Indeed, in addition to good cell-mediated immunity, it is precisely the inflammation, that plays an important role in the pathogenesis of gliomas [51].
Finally, because of a low number of patients, it is questionable to draw any conclusion. Present works did not receive a high enough priority, it is suggested to supplement the sample size to enrich the content of this article.
Asnwer:
- We deleted the final sentence from discussion: „However, taking into account both actual knowledge and our results, we suppose that HLA-G might be a promising relevant target for cancer immunotherapy.”
- And we shifted the sentence „HLA-G plays a great role in immunopathogenesis of gliomas and the plasma level of sHLA-G might have an impact on survival of glioma patients.” from „Conclusion“ to „Discussion“ with some changes: „HLA-G plays a great role in immunopathogenesis of cancer, and probably also gliomas, and the plasma level of sHLA-G might have an impact on survival of glioma patients.” The role of HLA-G in tumors is known, there is only lack of information concerning gliomas.
Thank you very much for your comments.
Best regards
Bucova Maria

Round 2
Reviewer 2 Report
I suggest that the manuscript is acceptable for the V2 version.
This manuscript is a resubmission of an earlier submission. The following is a list of the peer review reports and author responses from that submission.
Round 1
Reviewer 1 Report
The manuscript by Bucova and others reports a correlation between the expression of soluble HLA-G and glioma in a group of 59 patients at different stage of the disease compared to a cohort of 159 controls.
Major criticisms:
The results, differently from another study (ref 61), show no association between the disease and the insertion/deletion (del 14bp) at 3’-end of the gene affecting HLA-G expression. The plasma level of sHLA-G appears instead to be higher in patients. However, the control group in this case is restricted to 43 individuals. Why 43 out of 159? How these controls were selected? Since the test is barely significant (p:0.048), this becomes a crucial point.
The level of sHLA-G does not correlate neither with the polymorphism or with the disease stage but it weakly correlates with the status of the promoter methylation (which is not unexpected!). However, it is not said how the test was performed and from which samples (no description in MM!), Is it from the blood? In this case why it has not be done in the controls as well?
The correlation of sHLA-G with the IL-6 level performed in a cohort of 32 patients appears statistically more significant but again this has not been done in the cohort of controls.
Finally, the correlation with the surviving time of the patients is questionable. With such a low number of patients and so many parameters potentially affecting the results, is hazardous to make any conclusion.
Author Response
Dear reviewer,
we greatly appreciate your helpful comments and suggestions of our manuscript entitled: HLA-G 14bp ins/del Polymorphism, Plasma Level of Soluble HLA-G and Association with IL-6/IL-10 Ratio and Survival of Glioma Patients (Diagnostics-1517085). We have accepted all the recommendations to revise and improve the manuscript. The following revisions and changes were done:
Reviewer 1:
The manuscript by Bucova and others reports a correlation between the expression of soluble HLA-G and glioma in a group of 59 patients at different stage of the disease compared to a cohort of 159 controls.
Major criticisms:
- The results, differently from another study (ref 61), show no association between the disease and the insertion/deletion (del 14bp) at 3’-end of the gene affecting HLA-G expression. The plasma level of sHLA-G appears instead to be higher in patients. However, the control group in this case is restricted to 43 individuals. Why 43 out of 159? How these controls were selected? Since the test is barely significant (p:0.048), this becomes a crucial point.
The level of sHLA-G does not correlate neither with the polymorphism or with the disease stage but it weakly correlates with the status of the promoter methylation (which is not unexpected!). However, it is not said how the test was performed and from which samples (no description in MM!), Is it from the blood? In this case why it has not be done in the controls as well?
The correlation of sHLA-G with the IL-6 level performed in a cohort of 32 patients appears statistically more significant but again this has not been done in the cohort of controls.
Finally, the correlation with the surviving time of the patients is questionable. With such a low number of patients and so many parameters potentially affecting the results, is hazardous to make any conclusion.
Dear reviewer, thank you very much for your comments, we try to answer them step by step. The underlined sentences have been added to the manuscript.
Responses to reviewer 1:
- Genotyping of 14 bp ins/del polymorphism in the 3'UTR region of HLA-G was performed in the group of 59 patients with brain gliomas and 159 controls to increase the statistical power of analyzed values. We found a significantly higher proportion of glioma patients carrying the 14 nt insert in both homozygous and heterozygous states (14ins/ins and 14del/ins) compared to the control group of healthy subjects (P = 0.03). The study of Magalhães et al. (ref. 61, after adding citation for method for MGMT determination finally 62) found an association of the heterozygous 14bp Ins/Del genotypes with glioma of grade IV, while our study revealed an association with all group of glioma patients regardless the gliomas grading.
- The soluble HLA-G levels were analyzed in plasma of 59 patients with brain gliomas and 43 healthy controls. The selection criteria for the control group were plasma availability of healthy subjects that fulfilled the criteria of the absence of brain tumors or other malignancies, also the absence of autoimmune diseases, allergies (also in the family), and were without any signs of acute inflammation. These inclusion criteria were added to section “Subjects and Methods” in the manuscript.
Both Magalhães´s (studied gliomas grade IV) and our study revealed higher levels of soluble HLA-G – we, in the group of all glioma patients (with p-value = 0.048), Magalhães et al. (2021) in gliomas of grade IV – the worst grade of gliomas. As we can see (from Table 4), the plasma levels of sHLA-G increased in patients with glioma gradings, probably with an increased number of gliomas of grade IV, the p-value would be more significant also in our study. (it was added to the text as the 5th paragraph of “Discussion)
- The methylation analysis of the MGMT promoter was performed by a pathologist in the tumor tissue by bisulfite conversion-specific and methylation-specific PCR so that it can not be examined in healthy controls (Sasaki et al., Biochem Biophys Res Commun. 2003, 19;309(2):305-9). This method with citation has been added to the section “Subjects and Methods” in the manuscript and also as a citation. Our findings concerning the association of the levels of sHLA-G with MGMT methylation status have not been described so far. So, we described it as first.
- Since inflammation plays an important role in the pathogenesis of gliomas, we performed an association analysis of soluble HLA-G plasma levels with the levels of pro-inflammatory cytokine IL-6 in glioma patients. Our results showed, that the levels of sHLA-G negatively highly significantly correlated with the pro-inflammatory cytokine IL-6 concentration (Table 7, P = 0.0004). That means that sHLA-G exerts a strong anti-inflammatory activity. We were interested in the association of the level of sHLA-G with predominantly pro-inflammatory cytokine IL-6 in glioma patients, later we decided to investigated also IL-10, in order to find the inflammatory state. The P-value = 0.045 concerned the association with anti-inflammatory IL-10 and pro-inflammatory IL-6 ratio (IL-10/IL-6). The determination of pro-inflammatory cytokine IL-6 in healthy controls was not the subject of our study. Moreover, the serum/plasma levels of IL-6 in healthy subjects are very low, sometimes even beyond the detection limit (Kleiner G et al. Cytokine Levels in the Serum of Healthy Subjects. Mediators of Inflammation. Volume 2013, Article ID 434010, 6 pages (figure 3); Hyun Ok Kim. Serum cytokine profiles in healthy young and elderly populations assessed using multiplexed bead-based immunoassays. Journal of Translational Medicine 2011, 9:113).
- We are aware of a limited number of glioma patients, and recognize that there are many factors that can affect this survival, in our study we focused only on the HLA-G. (this was added We have precise this correlation, dividing patients into two groups: 1. With less and with more than 40 U/ml of sHLA-G. These groups differed also in one-year survival. We do not draw comprehensive conclusions about the survival, and we realize that a larger patient and control group would have greater statistical power, however, the number of study subjects was limited by their availability. We assume our present-day results as useful for future investigations on this topic. (this part was added at the end of the discussion)
We are aware of a limited number of glioma patients, and recognize that there are many factors that can affect this survival, in our study we focused only on the HLA-G. The levels of sHLA-G highly negatively correlated with the concentration of pro-inflammatory cytokine IL-6 and positively with IL-10/IL-6 ratio in plasma of glioma patients. HLA-G plays a great role in the immunopathogenesis of gliomas and the plasma level of sHLA-G might have an impact on the survival of glioma patients. This last paragraph was added to the conclusion.
Reviewer 2 Report
The manuscript from Maria Bucova and colaborators determined the HLA-G 14 bp ins/del polymorphism and the plasma level of soluble HLA-G as well as IL-6 and IL-10 in glioma patients. They identified that HLA-G 14 bp insert is more frequent in glioma patients and low soluble HLA-G levels are associated with longer overall survival. Additionally, the IL-6 levels and IL-10/IL-6 ratio is negatively associated with soluble HLA-G levels. These findings are interesting, since the evaluation of soluble HLA-G, IL-6 and IL-10 levels could be a future prognostic tool for glioma patients.
However, the quality of the determined data has to be evaluated and reviewed, since primer/probe-sets and ELISAs can be unprecise for the determination of the HLA-G 14 bp ins/del and soluble HLA-G plasma levels.
Author Response
Dear reviewer,
we greatly appreciate your helpful comments and suggestions of our manuscript entitled: HLA-G 14bp ins/del Polymorphism, Plasma Level of Soluble HLA-G and Association with IL-6/IL-10 Ratio and Survival of Glioma Patients (Diagnostics-1517085). We have accepted all the recommendations to revise and improve the manuscript. The following revisions and changes were done:
Reviewer 2:
The manuscript from Maria Bucova and collaborators determined the HLA-G 14 bp ins/del polymorphism and the plasma level of soluble HLA-G as well as IL-6 and IL-10 in glioma patients. They identified that HLA-G 14 bp insert is more frequent in glioma patients and low soluble HLA-G levels are associated with longer overall survival. Additionally, the IL-6 levels and IL-10/IL-6 ratio is negatively associated with soluble HLA-G levels. These findings are interesting since the evaluation of soluble HLA-G, IL-6, and IL-10 levels could be a future prognostic tool for glioma patients.
However, the quality of the determined data has to be evaluated and reviewed, since primer/probe-sets and ELISAs can be unprecise for the determination of the HLA-G 14 bp ins/del and soluble HLA-G plasma levels.
Response to reviewer 2:
- HLA-G 14 bp ins/del polymorphism is localized at the 3'UTR region and is the most studied polymorphism of the HLA gene. This polymorphism was genotyped by polymerase chain reaction (PCR) as described by Hviid et al. (2002). DNA was amplified by forward primer 5′GTGATGGGCTGTTTAAAGTGTCACC-3′ and reverse primer 5′GGAAGGAATGCAGTTCAGCATGA-3′ using a PCR cycler. PCR fragments of 224 bp (14 bp insertion) and PCR fragments of 210 bp (14 bp deletion) were identified. These primer sets were used for genotyping of HLA-G 14 bp ins/del polymorphism in other groups of patients, e.g. kidney transplant recipients and pre-eclampsia patients, as was reported in our previous studies (Durmanova et al., Open Life Sci 2016, 11: 372-379; Durmanova et al., BMJ 2017, 118: 517-522).
Soluble HLA-G plasma levels were determined by a commercial human sHLA-G ELISA kit (Exbio, BioVendor, Czech Republic). This ELISA kit is responsible for the analysis of soluble HLA-G molecules, namely shed HLA-G1 and secreted HLA-G5. The limit of sensitivity was 3 U/ml. About 100 U sHLA-G corresponds to 40–50 ng of protein. The ELISA kit was also used for the analysis of soluble HLA-G levels in kidney transplant recipients as previously reported (Durmanova et al., Open Life Sci 2016, 11: 372-379).
